# Application of Real and Virtual Radial Arm Maze Task in Human

**DOI:** 10.3390/brainsci12040468

**Published:** 2022-03-31

**Authors:** Tommaso Palombi, Laura Mandolesi, Fabio Alivernini, Andrea Chirico, Fabio Lucidi

**Affiliations:** 1Department of Social and Developmental Psychology, “Sapienza” University of Rome, 00185 Rome, Italy; fabio.alivernini@uniroma1.it (F.A.); andrea.chirico@uniroma1.it (A.C.); fabio.lucidi@uniroma1.it (F.L.); 2Department of Humanities, University of Naples “Federico II”, 80133 Naples, Italy; laura.mandolesi@unina.it

**Keywords:** human navigation, virtual reality, behavioral task, spatial abilities, large-scale task

## Abstract

Virtual Reality (VR) emerges as a promising technology capable of creating different scenarios in which the body, environment, and brain are closely related, proving enhancements in the diagnosis and treatment of several spatial memory deficits. In recent years, human spatial navigation has increasingly been studied in interactive virtual environments. However, navigational tasks are still not completely adapted in immersive 3D VR systems. We stipulate that an immersive Radial Arm Maze (RAM) is an excellent instrument, allowing the participants to be physically active within the maze exactly as in the walking RAM version in reality modality. RAM is a behavioral ecological task that allows the analyses of different facets of spatial memory, distinguishing declarative components from procedural ones. In addition to describing the characteristics of RAM, we will also analyze studies in which RAM has been used in virtual modality to provide suggestions into RAM building in immersive modality.

## 1. Introduction

Virtual reality (VR) has been traditionally defined as “interactive, virtual image displays enhanced by special processing and by non-visual display modalities … to convince users that they are immersed in a synthetic space” [1]. Nowadays, VR emerges as a promising technology useful in different fields of clinical applications [2,3]. In particular, by means of creating different scenarios in which the body, environment, and brain are closely related, VR facilitates the improvement and/or the recovery of spatial cognitive abilities, thus offering a great advantage in diagnosis and treatment of several spatial memory deficits [4,5,6,7,8,9]. The key strength of VR for screening spatial abilities refers to the capacity to mimic real-world tasks showing a correlation between real and virtual environments [10]. The simulated environment can be easily manipulated, facilitating experimental tasks that are difficult to implement in real-world settings. Other benefits of VR regard the possibility to participate in potentially dangerous tasks, such as moving in a complex environment or driving a car, in a controlled ecological setting [11], thus allowing to perform a task safely [12,13,14,15,16].

Several clinical evidences had documented the use of VR for screening spatial memory impairments in patients with Mild Cognitive Impairment (MCI) and Alzheimer Disease (AD) [3,11,14,17,18,19,20]. Moreover, VR technology had been implemented in the rehabilitation training of spatial abilities on patients with cognitive-spatial impairments, thus demonstrating how it represents valid support to the main methods of intervention [21].

The technological progress of VR has expanded the range of tools and types of questions on spatial learning, adapting most of the standard spatial tasks to the virtual version. The Radial Arm Maze (RAM), developed by Olton and Samuelson (1976) [22] and quickly adapted in humans, is a high ecological spatial task, firstly used in a real environment and subsequently in the virtual one.

The continuous evolution of VR devices has brought new challenges for researchers who implemented the standard tools to assess spatial abilities with new technological devices. Focusing on the RAM task, it is necessary to carry out an overview to understand how technology has necessitated an adaptation of RAM used for screening and rehabilitation of spatial abilities.

The purpose of the present study is to narratively review the literature dealing with the RAM task, both in its classic and virtual adaptations. With this aim, firstly, the traditional RAM paradigms and some of their applications in different clinical settings will be illustrated. Subsequently, the studies that have used the RAM task in virtual environments will be discussed to provide a careful reflection as to the potential and development of completely immersive RAM versions, which are still little used.

## 2. The Evolution of Virtual Reality Technology

The concept of VR was formulated in 1960 by computer scientist Ivan Sutherland, who created the Ultimate Display, the first head-mounted device. Although several definitions of VR have been formulated, the main features of the VR system are the immersion, the sense of the presence in a simulated environment, and the interaction with that environment [23,24,25]. In particular, the immersion depends on the characteristics of the VR system to simulate real interactions and stimuli in a virtual environment [25]. In scientific literature are reported three different types of VR systems that provide different degrees of immersion:

Non-immersive VR systems include the development of a 2D virtual environment projected on a computer screen, such as serious videogames. In this condition, using a PC monitor, keyboard, and a mouse, but also joysticks or gamepads, the subject performing a spatial task can be forced to view the scene using a route perspective [26].

Immersive VR systems provide a complete simulated experience using several devices, such as head-mounted displays, audio, and haptic devices. This technology enhances the stereoscopic view of the 3D simulated environment through the movement of the user’s head [27], and offers 360° audio and visual stimuli to perceive the user’s interaction as real [28,29].

Semi-immersive VR systems have some features in common with the two systems mentioned above. This technology includes a stereo image of a 3D simulated environment viewed on a PC monitor using a perspective projection linked to the head position of the observer [27,30].

Other features of VR, as mentioned before, regard the sense of presence and interaction with the virtual environment. The sense of presence regards the sensation and feelings of the users to be physically present in the virtual environment, having the opportunity to interact and react with the stimuli as if the user was in the real world [27].

Although the concept of VR was formulated in 1960, over the last 25 years, several scholars explored the utility, effects, and applications of this technology [27]. In the first phase, studies of VR referred to the computer graphics field. Successively, the researchers extended VR technology to several disciplines, representing a valuable tool for psychologists and neuroscientists [27].

In most recent years, the amount of research into psychological health and neuroscience has increased, demonstrating the validity of the VR tools. A recent review conducted by Cipresso and colleagues showed that VR had been widely used in navigation studies [27]. The Nobel prize winner Edvard Moser highlighted the potential of VR that can be implemented in navigational studies [31]. This evidence shows that the development and implementation of VR systems in research and clinical practice could provide a valuable contribution to assessing spatial abilities.

## 3. Main Spatial Tasks Suitable for Virtual Environments

Experimental psychological research offers several navigational tasks to study spatial abilities. Nevertheless, few studies adapted the classical spatial experimental paradigms using VR technology. In this contest, spatial cognition research has distinguished itself for developing the virtual versions of the Morris Water Maze (MWM) and the Radial Arm Maze (RAM), the behavioral tasks most used in the study of spatial function [32,33,34,35]. Both tasks were used in the animal cognition field. Only in most recent years, they are becoming part of the evaluation tools in research involving human participants to study the development of spatial abilities, or their functioning in specific clinical populations [13,36,37,38,39].

The MWM adapted to humans consists of a large circular pool in which the subject has to find a hidden object, while the RAM is formed by a central holding area from which a number of identical arms radiate. Although both tasks have a unique contribution in analyzing spatial memory, compared to MWM, RAM offers the great advantage of being administered with several paradigms that investigate the different faces of the spatial memory process, providing more opportunities to evaluate the spatial working memory component compared to the real and virtual pool [35]. In addition, while in the MWM task, the subject may perform an infinite number of path trajectories, whereas in the RAM task, the choices of the entries are constrained, thus facilitating the identification of the strategy put into action. Moreover, thanks to the many parameters that can be calculated, RAM is a sensitive and ecological tool for the diagnosis and study of spatial deficit. However, it should be noted that many studies analyzing spatial abilities in humans have used new apparatuses including both MWM and RAM tasks, i.e., [36] have developed new navigational paradigms to compare the spatial abilities in real and virtual modality [37] alongside other research.

Despite these advantages, the use of RAM in VR technology is still too limited and not completely adapted in immersive VR systems.

## 4. Method

A comprehensive search within the literature was used to detect the available studies related to the use of RAM in the real and virtual modalities in the human sample. We carried out searches in PubMed/Medline, Scopus, Web of Science, and Pubmed databases in the date range from 1976 (date of the first RAM study) to December 2021. Search terms included a combination of “Radial Maze”, “Arm Maze”, “Radial Arm Maze”. To avoid animal studies, articles containing the following keywords have been excluded: “Mice”, “Rat”, “Rodent”, “Gerbil”, “Pigeon”, “Fish”. Once the main articles were identified, to supplement the aforementioned search terms, a second search was carried out using the citations within each article.

Only articles in peer-reviewed journals and written in the English language were included.

Studies that arose from the search terms were assessed for further evaluation via abstract review, and duplicates were removed. In addition, the reference lists of the full-text articles were manually checked to identify other studies that were cross-referenced, in order to find further existing articles on the use of RAM. As a result, we detected in total 433 potentially relevant articles. After screening the titles and abstracts, a total of 352 articles were eliminated. The remaining 81 studies were examined by full-text review, and 16 of them were further excluded until they reached 65 studies in total. All the excluded studies did not meet the eligibility criteria. Figure 1 is reported the PRISMA flow diagram regarding the search process [38].

The high number of studies excluded in the screening phase is due to the broad use of RAM in animal research. The eligibility criteria referred to spatial abilities being evaluated using RAM. Specifically, we have selected only studies covering the different facets of spatial memory processes. The 16 studies were excluded in the eligibility phase because they focused on other processes, such as attention or perception. Various data were extracted from each study, including the number and age of participants, type of population, RAM paradigm, and real or virtual environment (Table 1 and Table 2).

## 5. Radial Arm Maze Task (RAM)

### 5.1. Free-Choice and Forced-Choice Version

The Radial Arm Maze (RAM) task, developed by Olton and Samuelson (1976) to assess the spatial abilities in rodents, is also used in several studies on children [42,45,46,47,51,55] and adults [48,49,50].

RAM consists of a central holding area from which several identical arms, commonly eight, radiate, and the task’s difficulty depends on the number of them. There is a hidden reward at the end of each arm, generally a coin or a little toy for children.

Different RAM paradigms take into account the environmental cues and the kind of spatial memory process to be investigated. Generally, it is possible to distinguish two main classic paradigms: free-choice and forced-choice RAM versions (Figure 2).

In the free-choice version, the subjects have to take all the rewards and know that the arms are rewarded only once (declarative rule). To solve the task without errors, the subject has to make use of mnesic and mapping abilities, as well as proficient explorative strategies [46,98,101]. Carrying out the RAM in several trials, these competencies can be learned during the various phases of the task.

For many years, the free-choice RAM version has been considered appropriate for evaluating the correct functioning of working-term memory abilities by detecting the number of errors (e.g., returning to arms already visited). However, some authors observed that the longest sequence of correctly visited arms, corresponding to spatial span parameter, can also depend on the type of strategy put into action to explore the maze, suggesting the potential that RAM offers to evaluate procedural memory processes [102,103]. In addition, employing different parameters, the free-choice RAM version allows to efficaciously study the explorative strategies used by the subject. For example, it is possible to analyze if he/she visits a specific sequence of arms or always beginning a run from the same arm (“praxic” strategy), or if he/she solves the task by referring to specific environmental stimuli (“taxic” strategy), or finally, if he/she exploits mapping abilities to build a cognitive spatial map (“place” strategy) [53,101,104]. To refer to these different strategies, numerous terms, sich as “motor”, “cue”, or “relational” strategies, respectively, have been used in other studies [46,101].

Therefore, to distinguish explorative from working mnesic components, it is possible to use the forced-choice RAM version. In this protocol, each trial consists of two phases. In the first phase, the subjects have to collect only four rewards, while the remaining ones are inaccessible. In the second phase, he/she has to collect the rewards of the four arms not visited in the first phase. Success depends on remembering the arms visited in the first phase (rather than putting into action particular search patterns), thus emphasizing working memory requirements. Putting into action a specific exploratory strategy is avoided using different angles to separate the opened arms (i.e., arms 2,4,5,8). Therefore, although both paradigms investigate spatial memory processes, they analyze different aspects of these processes.

Both versions of RAM can be cued or uncued [46,56]. In the cued RAM version, each arm is made physically distinct by visual stimulus at its end. In the uncued RAM version, visited arms can be remembered by the subjects in relation to their spatial relationship to distal extra maze cues. It is easy to understand that when using the cued RAM version, the subject is forced to apply a taxic strategy to solve the task, while an allocentric strategy in the uncued RAM version is more appropriate. Again, the type of paradigm allows to investigate different aspects of spatial exploration.

Therefore, the choice of a specific RAM paradigm depends on the type of study objective to be achieved, and the normotypical and clinical population to be studied. For example, for children around the age of four years, who have not yet developed short-term memory processes, the use of free choice may be more appropriate.

In Table 3, the main parameters analyzed in free-choice and forced-choice RAM versions are reported to emphasize the different faces of memory components that can be studied throughout real and virtual RAM tasks.

It is appropriate to note that in free-choice and forced-choice RAM versions, the subject walks around the maze, and this promotes the integration of the mechanisms that link perception to action. This feature suggests that RAM is a complete task as it allows to also analyze perceptive and motor processes. Furthermore, the exploration of an environment through moving in it accelerates the spatial learning processes, allowing the formation of a spatial cognitive map [105], thus indicating RAM as a tool for devising virtual personalized neurorehabilitation training, as is already being done with other experimental protocols [106].

### 5.2. Table RAM and Visuospatial Peripersonal Abilities

The RAM is cataloged among large-scale behavioral tasks since it is a walking task. The subjects are inside the maze and see it from the inside, thus promoting an allocentric and egocentric encoding. The participant is compelled to build a spatial cognitive map of RAM to orient and move himself/herself in it. In this way, the declarative competence of the environment is probably built through procedural competence [103]. Recently, Foti and collaborators have developed a RAM table version that allows studying the visuospatial peripersonal abilities through body–objects interaction [54]. In fact, in this table RAM version, the participant is forced to explore the portion of space accessible with the limbs in order to resolve the task, which was presented to children as the “Ladybug game”. The child had to move the older sister ladybug, placed on the central platform, to find its sisters hidden inside the caps at the end of each arm [54]. The child is seated in front of the RAM and has visual access to the maze in all its completeness. Seeing it from above, it is likely that the construction of the spatial cognitive map may be facilitated because declarative knowledge is promptly formed. In addition, recent scientific literature reported that tactile and visual stimuli inside the peripersonal space elicit stronger processing and induce a powerful multilevel activation [107], inducing an integration of perceptive, motor, and cognitive processes. When we see an object and recognize its function, we also know how to grasp it, preparing ourselves for the action to be enacted upon it. These characteristics related to the process that links the perception to the action suggest the table RAM is an advantageous tool to improve peripersonal spatial abilities. Furthermore, in this RAM table version, the two RAM paradigms, free choice and forced choice, were administered. However, this time it is necessary to point out that the free choice paradigm served as habituation to the setting. In contrast, the forced choice paradigm constituted the experimental part of the study. The reason for this is easy to understand, as on a small scale, free exploration is elementary, even for children. In the future, it may be helpful to administer free choice to populations with marked cognitive deficits, such as in neglect syndrome.

In this line of thinking, it is interesting to note that another group of researchers has used a small-scale RAM model to investigate the age at which children begin to integrate the increasing flexibility in the conjoint use of egocentric and allocentric frames of reference [41], obtaining data comparable to those of classic neuropsychological spatial tests, such as Corsi Block task or block construction [42,53], indicating also the reliability of this ecological task. In the past, O’Connor and Glassman used a radial maze analog drawn on paper to study short-term memory [48], first suggesting the RAM as a tabletop tool.

## 6. Applications of RAM Task in Real Environment

As described above, RAM is a behavioral ecological task on a large and small scale that allows the analyses of different facets of spatial memory. In humans, several clinical and psychological studies have extensively used the walking RAM version for analyzing the navigational abilities in individuals with typical development (TD) and the spatial deficit in specific clinical populations.

In the late ‘80s, walking RAM was used in children to study spatial memory and understand from what age it could be administered [45,47]. These studies have shown that even preschool infants can walk in RAM. However, the variable dimensions regarding length and number of arms and the experimental setting have confused the results. About ten years after these pioneering studies, Overmann and colleagues developed a RAM built to human scale in which children were tested without explicit verbal instructions and with a longitudinal procedure for up to 16 consecutive weekdays, using free-choice and forced-choice versions [46]. In a sense, Overmann’s study confirms the precedents, even though it aimed to observe the development of mapping abilities rather than evaluate the age of administration of RAM. Successively, other behavioral studies on TD children were carried out employing both versions of the walking RAM to investigate the ontogenesis of spatial competencies and eventually gender differences [42,43,44,55] as well, so as to better characterize the spatial deficit in adolescents with Williams and Prader-Willi syndromes [52,53] and to evaluate the spatial orientation of intrauterine growth retarded children [51]. All these studies have shown how RAM can analyze the development of a process (spatial abilities) and highlight the presence and severity of a spatial deficit.

Recently, the walking RAM task has also been used to compare learning by observation to learning by doing in TD children [40]. In this study, the authors have made clear that the observation of the correct explorative strategy showed by the experimenter promotes the development of spatial declarative and procedural knowledge, thus suggesting the RAM task as a useful tool for improving and facilitating spatial memory. In particular, the authors highlighted that the observation of a correct exploration strategy, such as the entry into the adjacent arms, induces an early development of the spatial cognitive map in the observing child. This study suggests that RAM can also be an educational tool to facilitate and accelerate learning processes.

Even in adults, the first studies that used the walking RAM date back to the 1980s. In some of these, participants’ performances were compared to those of the rats in analog mazes [48]. Successively, the RAM task was mainly used to study human navigation behavior in health and clinical populations [48,49,50,55,56,57], highlighting once again how RAM can be used for diagnostic purposes. Recently, a RAM version has been also used to evidence physical activity effects on spatial abilities [39]. In fact, by comparing the performance of athletes with those of a sedentary group, it was possible to highlight how physical exercise improves spatial memory.

However, in these studies, the behavioral procedure is not always comparable. For example, in some of them, it is preferred to use the free choice version with only part of the arms baited [96], or to insert specific cue intramaze, or change the starting arm [56]. As already pointed out, the choice of one or the other version of the RAM task depends on the age of the participants and on the type of memory process to be studied.

Although these differences make the results confusing and not homogeneous, they demonstrate once again the extent to which RAM task is a flexible tool that can be easily adapted to the type of spatial process to be investigated and the type of deficit to be rehabilitated.

## 7. Potentiality and Applications of RAM Task in Virtual Environment

The RAM task is a highly ecological test because it is administered outside hospital environments and experimental settings of research laboratories. Aside for a few examples, RAM is a large-scale task that is presented as a game, especially in children. When considering the different RAM paradigms and versions, overall, on the one hand, they have favored objectives and reliable results, also correlating to aseptic paper and pencil tests. However, on the other hand, their design has hampered RAM use, as it is very expensive to assemble them in real environments. Furthermore, as RAM tasks are very often performed outdoors and generally last a few days, they are also affected by weather conditions. All these difficulties may explain why the RAM task is only partially used in humans compared to its extensive application in animal research and the numerous evidences in the implementation in virtual modality (Table 2) made possible by VR technology progress. As has been already pointed out, VR offers several advantages, such as the possibility to evaluate people in complete safety [11,12]. Another possible advantage consists of manipulating the environment, for example, making it increasingly complex or easier to explore, thus allowing for more personalization as well as a more interactive subject-environment. In addition, the changes that can be made in virtual modality allow to specifically investigate the type of strategy used by the participant to solve the task. While in real RAM version, for example, it is not certain whether the participant has oriented himself/herself according to the external cues, which, although kept under control cannot be stable (for example, a strong wind, variable brightness, etc.). In virtual RAM version, it is possible to modify the surrounding landmarks and keep other conditions constant, analyzing the procedural competences in more detail. Despite this, most of the studies that have used RAM in virtual modality have adopted the forced-choice version of the task (Table 2), which allows analyzing working and short-term memory processes rather than the type of strategy used by the subject. A possible explanation could be that the free choice version is apparently easier than the forced-choice one, and since the participants were mainly young adults, the researchers believed it more useful to administer a RAM version emphasizing working memory requirements.

Other potential advantages relate to the fact that the digitized versions of the RAM task can be easily shared by several groups of researchers, and that the data obtained can be entered into scientific databases. With the aim of eventually implementing rehabilitative intervention, it could be possible to imagine a sort of “videogame training” that the patient can perform inside his/her home when motivated to do so.

The present review has evidenced 46 papers concerning virtual RAM task as the method chosen to investigate different aspects of human behavior, not only related to analyzing the spatial abilities. In most of them, a non-immersive VR modality of RAM task has been used.

To our knowledge, the first evidence of a virtual RAM task goes back to the Iaria et al. study in 2003. The authors created an eight-arm radial maze with a central starting location. The maze was surrounded by a landscape (mountains and sunset), two trees, and a short wall located between the landscape and the tree. At the end of each arm, there was a staircase leading to the location where an object could be picked up in some of the arms. The participants were young, healthy adults who used a keypad to move in any direction [34]. Successively, joysticks were also used to navigate through the virtual RAM, but the subjects were always seated in front of the computer [33,91,92,93,94,96,97,98,99,100]. In 2012, a study of 599, including TD children and younger to older healthy adults, demonstrated the virtual RAM task to be a useful tool with which to investigate the changes in exploratory strategies over life span [87], confirming the results of the studies conducted with real RAM, but adding valuable information as to environmental factors that can modulate the development of navigational strategies. In the last twenty years, studies with virtual RAM tasks have greatly increased, and more and more evidence also relates to clinical populations and children, as well as the suggestion of new paradigms of task-based RAM [74,75,76,77,78,79,80,81,83,88,94,95,96]. For example, Marsh and colleagues (2015) had administered a virtual eight RAM version during fMRI scanning of adults with obsessive-compulsive disorder (OCD) in order to study the functioning of mesolimbic and striatal areas involved in reward-based spatial learning [79]. Furthermore, other authors investigated navigational strategies in Attention Deficit Hyperactivity Disorder (ADHD) children [78]. The use of virtual RAM in these clinical populations suggests how it is suitable for individuals exhibiting behavioral alterations.

Recently, a digitalized version of the RAM task was also used to investigate the impact of the COVID-19 pandemic on the spatial exploration in Italian University students, allowing to evidence an increase in pseudoneglect through analysis of the lateralization of the first explored arm [60]. However, many studies use RAM in non-immersive modality to evaluate spatial abilities [61,62,63,64,65,66,67,68,69,70,71,72,73].

To date, only two studies reported the virtual RAM task in full immersive modality [58,59]. In particular, Kim et al. have developed a virtual RAM task with a head-mounted display to produce information about travel distance and head movement, demonstrating that this virtual task was just as competent as the walking task one in measuring spatial learning and memory [58]. More recently, Ben-Zeev and colleagues have produced a virtual RAM task in which the subjects wore specific virtual reality goggles as a display that enabled them to see the room in a first-person perspective, as well as a rotating tool of the view, due to its capability to translate head movements in real-time as shifts of the viewpoint [59].

From the analysis of the studies carried out, it is clear that most of them use the forced choice paradigm (Table 2), and this observation deserves careful consideration. Once again, the two paradigms allowed to evaluate different facets of spatial memory. Still, the forced choice method is more sensitive to the short memory components, and is also more challenging to perform. However, in virtual modality, it is easier to modify scenarios by reducing (or increasing) the complexity of the task. Perhaps this could be why forced choice in virtual modality is more frequent than free choice.

## 8. Conclusion and Future Perspectives

The present review highlights the potential of the RAM task to study spatial abilities in real and virtual environments. Furthermore, the analysis of the scientific literature has shown an increase in studies with virtual RAM tasks.

However, almost all of these tasks are based on non-immersive methodology that offers several advantages, but does have some limitations. It is important to underline the benefits of using non-immersive virtual RAM tasks, as the experimenter can easily modify the scenario. Moreover, using non-immersive VR, it is possible to build virtual environments that can be seen by the subject from a monitor, forcing both a view from above (i.e., the map of a city) and a view from the inside (i.e., using a third-person perspective in which the subject sees his/her or another virtual body moving in a videogame), promoting thus the building of the spatial cognitive map influencing egocentric and allocentric encoding [63].

In relation to limits that non-immersive VR has, being a simple video viewing experience, it cannot permit the experience of feeling like an integral part of a context [28]. Thus, RAM exploration is not entirely comparable to real exploration in the maze, and care should be taken when evaluating patients with a spatial deficit [10]. However, these limits can be overcome using immersive VR. Compared to non-immersive VR, immersive VR leverages the vestibular and proprioceptive systems to further engage the participant, and this involvement seems to be related to spatial memory recall [108]. Recently, it has been demonstrated that walking induces oscillations, peaking within a lower frequency band in the hippocampus, which correlate to spatial memory processing [109], thus suggesting immersive VR is a suitable technology for promoting spatial memory. Therefore, using full immersive VR, it is possible to stress procedural and declarative components of spatial memory processes [82,84,110], which are equally necessary for efficient exploration and construction of the cognitive spatial map [42,85,101]. Just like in a real environment, through full immersive VR, the subject explores the virtual scenario considering the position of the virtual environmental cues and his/her own position to these (declarative competence). Simultaneously, he/she can move in the virtual environment to reach (or avoid) specific objects, putting into action an exploration strategy (procedural competence) [54]. These observations encourage careful consideration in promoting therapeutic strategies for improving and recovering spatial abilities through full immersive virtual navigation training, and in this context, the RAM task reveals itself to be a suitable tool for use in neurorehabilitation. However, it is important to emphasize that there are also other spatial tasks adapted in virtual modality, such as the virtual MWM or the virtual boxes room task [106,111], and that new research has to be conducted in order to find which procedure is more useful for the assessment and rehabilitation of spatial memory in different pathologies.

## Figures and Tables

**Figure 1 brainsci-12-00468-f001:**
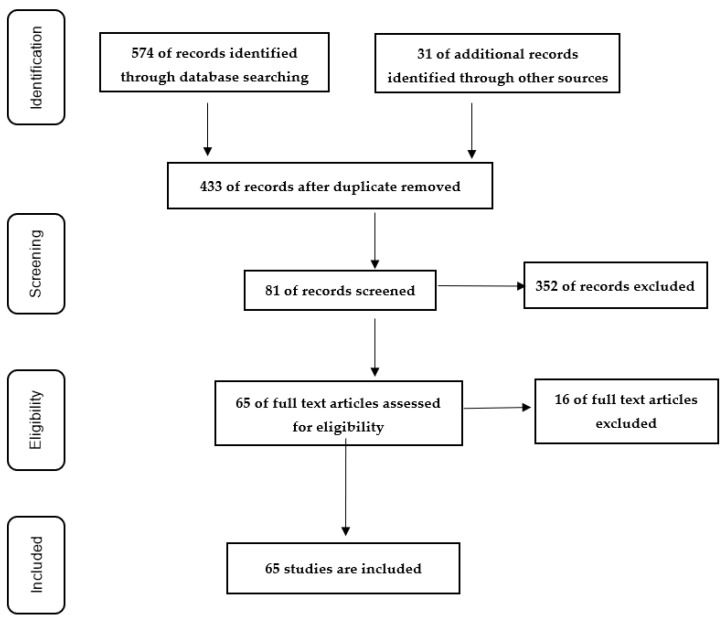
PRISMA flow diagram.

**Figure 2 brainsci-12-00468-f002:**
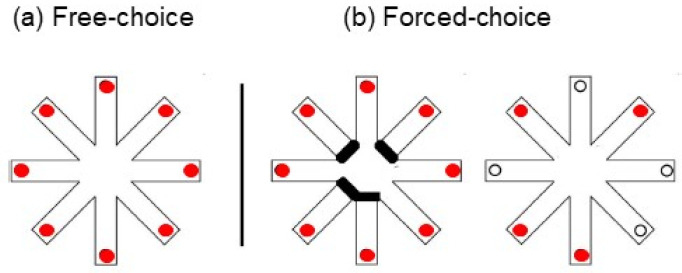
Free-choice (**a**) and forced-choice (**b**) paradigms are represented.

**Table 1 brainsci-12-00468-t001:** Studies carried out using RAM in real environment.

Authors	Sample Characteristics	RAM Paradigm	Arms
Serra et al., 2021 [39]	Children (healthy)	Forced-choice/ Table RAM	8
	N = 28; Age: m = 8.1, SD = 1.1		
Foti et al., 2018 [40]	Children (healthy)	Free-choice	8
	N = 36; Age: m = 5.3, SD = 0.2		
Moraleda et al., 2013 [41]	Children (healthy)	Free-choice/ Table RAM	4
	N *=* 48; Reported age range from 6 to 10 years		
Mandolesi et al., 2009 [42]	Children (healthy)	Free-choice Forced-choice	8
	N *=* 90; Reported age range from 3 to 8 years		
Foreman et al., 1994 [43]	Children (healthy)	Free-choice/ Forced-choice	12
	N *=* 28; Reported age: m = 6, SD = 4.5		
Foreman et al., 1990 [44]	Children (healthy)	Forced-choice	10
	N *=* 18; Age: m = 3.3, SD: Not reported		
Foreman et al., 1984 [45]	Children (healthy)	Free-choice	8
	N *=* 10; Reported age range from 2 to 4 years		
Overman et al., 1996 [46]	Children/young adults (healthy)	Free-choice/ Forced-choice	8
	N *=* 43; Children reported age range from 20 to 151 months; Young adults age range from 17 to 21 years		
Aadland et al., 1985 [47]	Children/young adults (healthy)	Free-choice/ Forced-choice	8
	N *=* 146; Reported age range from 18 to 71 months		
O’Connor & Glassman, 1993 [48]	Young adults (healthy)	Free-choice (drawn on paper)	17
	N *=* 15; Age: Not reported		
Glassman et al., 1994 [49]	Young adults (healthy)	Free-choice	17/13
	N *=* 57; Age: Not reported		
Glassman et al., 1998 [50]	Young adults/adults (healthy)	Free-choice	8
	N *=* 21; Reported age range from 18 to 35 years		
Leitner et al., 2005 [51]	Children (Intrauterine growth retardation)	Free-choice/ Forced-choice	8
	N *=* 28; Reported age: 6 years, SD: Not reported		
Foti et al., 2011 [52]	Children/adolescents (Prader-Willi syndrome and Williams syndrome)	Free-choice/ Forced-choice	8
	N *=* 24; PWS Mental age: m = 6.0, SD = 0.5; WS Mental age: m = 6.0, SD = 0.3		
Mandolesi et al., 2009 [53]	Children/Adolescents (Williams syndrome)	Free-choice/ Forced-choice	8
	N *=* 14; Mental age: m = 6.2, SD = 1.4		
Foti et al., 2020 [54]	Adolescents (Williams syndrome)	Free-choice/ Forced-choice	8
	N *=* 15; Mental age: m = 18.1, SD = 5.2		
Bertholet et al., 2015 [55]	Young adults (Intellectual disability and healthy)	Free-choice/ Forced-choice	8
	N *=* 107; Age: m = 22.8, SD = 0.7		
Palermo et al., 2014 [56]	Adults (Developmental topographical disorientation)	Free-choice/ Forced-choice	8
	N *=* 2; patient 1: reported age 29 years; patient 2: reported age 38 years		
Bohbot et al., 2002 [57]	Adults (Temporal lesions and healthy)	Free-choice	8
	N *=* 52; Age: m = 38, SD = 1.3		

Abbreviations: m, mean; SD, standard deviation; RAM, Radial Arm Maze; PWS, Prader-Willi Syndrome; WS, Williams Syndrome.

**Table 2 brainsci-12-00468-t002:** Studies carried out using RAM in virtual environment.

Authors	Sample Characteristics	Virtual Modality	RAM Paradigm	Arms
Kim et al., 2018 [58]	Young adults (healthy)	Immersive	Forced-choice	8
	N *=* 80; Age: m = 23.2 SD = 2.4			
Ben-Zeev et al., 2020 [59]	Young adults (healthy)	Immersive	Free-choice	8
	N *=* 40; Age: m = 25.5, SD = 1.6,			
Patel et al., 2021 [35]	Young adults (healthy)	Non-immersive	Forced-choice	8
	N *=* 86; Age: m = 19.0, SD = 1.0			
Somma et al., 2021 [60]	Young adults (healthy)	Non-immersive	Free-choice	8
	N *=* 47; Age: m = 20, SD = 1.0			
Taheri Gorji et al., 2021 [61]	Young adults (healthy)	Non-immersive	Forced-choice	6
	N *=* 42; Age: m = 24.4, SD = 2.4			
Rechtman et al., 2020 [62]	Children (exposed to manganese)	Non-immersive	Forced-choice	8
	N *=* 188; Age: m = 12.01, SD = 0.9			
Sodums & Bohbot, 2020 [63]	Elderly people (healthy)	Non-immersive	Forced-choice	12
	N *=* 39; Age: m = 64.6, SD = 4.1			
Dahmani et al., 2020 [64]	Young adults (healthy)	Non-immersive	Forced-choice	8
	N *=* 55; Age: m = 22.9, SD = 3.5			
Goodman et al., 2020 [65]	Young adults (healthy)	Non-immersive	Forced-choice	8
	N *=* 62; Age: m = 23, SD = 6.3			
Yang et al., 2019 [66]	Children/young adults (healthy)	Non-immersive	Free-choice	6
	N *=* 83; Children: reported age range from 6 to 10 years; Young adults: reported age range from 18 to 22 years			
Caplan et al., 2019 [67]	Young adults (healthy)	Non-immersive	Free-choice	8
	N *=* 173; Age: m = 19.5, SD = 2.5			
Aumont et al., 2019 [68]	Young adults (healthy)	Non-immersive	Forced-choice	8
	N *=* 50; Age: m = 23.4, SD = 4.1			
Aumont et al., 2019 [69]	Young adults (healthy)	Non-immersive	Forced-choice	8
	N *=* 53; Age: m = 23.9, SD = 4.4			
Raiesdana, 2018 [70]	Young adults (healthy)	Non-immersive	Free-choice	8
	N *=* 8; Age: m = 22.1, SD = 2.3			
Dahmani et al., 2018 [71]	Young adults/adult (healthy/focal lesion to the frontal lobe)	Non-immersive	Forced-choice	8
	N *=* 78; Young adults: Age: m = 22.9, SD = 3.5; Patients: group 1: Age m = 56.6, SD = 16.2; group 2: Age m = 60, SD = 6.7			
Aumont et al., 2018 [72]	Young adults (healthy)	Non-immersive	Forced-choice	8
	N *=* 50; Age: m = 23.4, SD = 4.1			
Konishi et al., 2018 [73]	Elderly people (healthy)	Non-immersive	Forced-choice	12
	N *=* 66; Age: m = 66.1, SD = 4.5			
Bauer et al., 2017 [74]	Children (exposed to manganese)	Non-immersive	Forced-choice	8
	N *=* 142; Age: m = 12.4, SD = 0.9			
Wilkins et al., 2017 [75]	Adults (Schizophrenia)	Non-immersive	Forced-choice	8
	N *=* 16; Age: m = 44.4, SD = 6.1;			
Cyr et al., 2016 [76]	Adolescents (Bulimia nervosa)	Non-immersive	Free-choice	8
	N *=* 27; Age: m = 16.6 SD = 1.5			
Migo et al., 2016 [77]	Elderly people (Mild cognitive impairment)	Non-immersive	Free-choice	8
	N *=* 8; Age: m = 69.6, SD = 5.8			
Robaey et al., 2016 [78]	Children (ADHD)	Non-immersive	Forced-choice	8
	N *=* 223; Age: m = 8.4, SD = 0.1			
Marsh et al., 2015 [79]	Adults (Obsessive compulsive disorder)	Non-immersive	Free-choice	8
	N *=* 33; Age: m = 29.4, SD = 8.1			
Lee et al., 2014 [80]	Elderly people (Alzheimer’s Disease, Mild cognitive impairment)	Non-immersive	Forced-choice	6
	N *=* 40; AD Age: m = 72.4, SD = 5.6; MCI: Age m = 70.7, SD = 5			
Pirogovsky et al., 2013 [81]	Elderly people (Mild Cognitiv Impairment)	Non-immersive	Free-choice	8
	N *=* 10; Age: m = 76.8, SD = 2.3			
Konishi & Bohbot, 2013 [82]	Elderly people (healthy)	Non-immersive	Forced-choice	8
	N *=* 45; Age: m = 64.3, SD = 4			
Wilkins et al., 2013 [83]	Adults (Schizophrenia)	Non-immersive	Forced-choice	8
	N *=* 17; Age: m = 44.9, SD = 4.8; Age: m = 39.6, SD = 9.8			
Konishi et al., 2013 [84]	Young adults/elderly people (healthy)	Non-immersive	Forced-choice	12
	N *=* 52; Young adults: Age: m = 23.8, SD = 3.8; Elderly people: Age: m = 64.2, SD = 4.7			
Andersen et al., 2012 [85]	Young adults (healthy)	Non-immersive	Forced-choice	8
	N *=* 7; Age: m = 28.1, SD = 5.6			
Braun et al., 2012 [86]	Children (exposed to manganese)	Non-immersive	Forced-choice	8
	N *=* 255; Age: m = 13, SD = 0.9			
Bohbot et al., 2012 [87]	Children/young adults/elderly people (healthy)	Non-immersive	Forced-choice	8
	N *=* 599; Children: Age: m = 8.4, SD = 0.1; Young adults: Age: m = 25.6, SD = 4.6; Elderly people: Age: m = 65.6, SD = 5.6			
Spieker et al., 2012 [88]	Adults (Schizophrenia)	Non-immersive	Forced-choice	8
	N *=* 33; Age: m = 40.0, SD = 11.9			
Schwabe et al., 2012 [89]	Young adults (healthy)	Non-immersive	Forced-choice	8
	N *=* 60; Age: m = 24.4, SD = 0.4			
Etchamendy et al., 2012 [90]	Young adults/elderly people (healthy)	Non-immersive	Forced-choice	12
	N *=* 55; Young adults: Age: m = 25.1, SD = 4.1; Elderly people: Age: m = 66.9, SD = 7.9			
Bohbot et al., 2011 [91]	Young adults (healthy)	Non-immersive	Forced-choice	8
	N *=* 66; Age: m = 21.6, SD = 0.81;			
Banner et al., 2011 [92]	Young adults (healthy)	Non-immersive	Forced-choice	8
	N *=* 106; Age: m = 23.4, SD = 1.1			
Marsh et al., 2010 [93]	Adults (healthy)	Non-immersive	Free-choice	8
	N *=* 25; Age: m = 32.5, SD = 7.6			
Goodrich-Hunsaker & Hopkins, 2010 [94]	Adults (Hippocampal damage)	Non-immersive	Forced-choice	8
	N *=* 5; Age: m = 45.6, SD = 9.4			
Pirogovsky et al., 2009 [95]	Adults (Huntington’s disease)	Non-immersive	Free-choice	8
	N *=* 18; Age: m = 46.4, SD = 2.1			
Rahman & Koerting, 2008 [96]	Young adults/adults (healthy)	Non-immersive	Forced-choice	8
	N *=* 140; Reported age range from 19 to 45 years			
Bohbot et al., 2007 [97]	Young adults (healthy)	Non-immersive	Forced-choice	8
	N *=* 30; Age: m = 27.9, SD = 4.1			
Levy et al., 2005 [98]	Young adults (healthy)	Non-immersive	Forced-choice	12
	N *=* 55; Reported age range from 18 to 30 years			
Astur et al., 2004 [99]	Young adults (healthy)	Non-immersive	Forced-choice	8
	N *=* 13; Reported age range from 18 to 30 years			
Astur et al., 2002 [100]	Young adults (healthy)	Non-immersive	Forced-choice	8
	N *=* 61; Age: m = 19.4, SD = 4.8			
Bohbot et al.,2004 [33]	Adults (medial temporal lobe resections)	Non-immersive	Forced-choice	8
	N *=* 15; Age: m = 42.5, SD = 8.7			
Iaria et al., 2003 [34]	Young adults (healthy)	Non-immersive	Forced-choice	8
	N *=* 50; Age: m = 27.7, SD = 4.7			

Abbreviations: m, mean; SD, standard deviation; RAM, Radial Arm Maze; MCI, Mild Cognitive Impairment; AD, Alzheimer’s Disease.

**Table 3 brainsci-12-00468-t003:** Illustration of the main parameters used to analyze the performances in RAM task.

Free-Choice RAM Version	Forced-Choice RAM Version *(Referred to the Second Phase of the Task)*
Total time to complete the entire task Time to reach each reward	Total time to complete the second phase of the task
Latency to select the first arm	Latency to select the first arm
Total entries (arms correct and incorrect visited)	Total entries (arms correct and incorrect visited)
Distance travelled	Distance travelled
Movement speed	Errors
Frequency of successes/Percentage of correct visits/Search efficiency	Across-phase errors
Errors/Error-free trials	Within-phase errors
The longest sequence of correctly visited arms	The longest sequence of correctly visited arms
Percentage of angles turned (45°, 90°, 135°, 180° or 360°)/Angle change/Strategy fixation	
Perseverations (consecutive entries into the same arm or the re-entries into a fixed sequence of arms)	
Declarative mastery	

## Data Availability

Data are available under request to the corresponding author Tommaso Palombi (tommaso.palombi@uniroma1.it).

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
