# Peer review of "Application of Real and Virtual Radial Arm Maze Task in Human"

_brainsci, 2022, doi:10.3390/brainsci12040468_

Round 1

Reviewer 1 Report

Review of Ms. brainsci-1581780, "An overview on application of real and virtual radial arm maze task" (sic!) by Tommaso Palombi, Laura Mandolesi, Fabio Aliverini, Andrea Chirico, and Fabio Lucidi

The authors aim to provide a review of research on the Radial Arm Maze (RAM) task in real and virtual environments, as well as a performance comparison between clinical and nonclinical populations in this task. The authors performed a literature search among more than 400 papers, homing in on a selection of 65 studies. Based on this selection, they describe some of the variations between tasks. To keep it short, the paper does not deliver what the abstract suggests and does not meet the necessary conditions for a review paper, neither in content nor in style.

MAJOR POINTS

- The title and abstract promise a very encompassing type of review: the RAM paradigm analyzed in real-world and virtual versions and in neurotypical and clinical populations. From the abstract, one would expect (1) a systematic comparison of research findings between real and virtual conditions, and (2) a systematic comparison between clinical and non-clinical populations with respect to important task parameters. Instead, the paper only describes variations of the paradigm and its behavioral parameters, albeit without going into enough technical detail to make this description practically useful. It fails to describe or even mention the most important results coming from this type of research, and it never compares findings from different methodologies as promised in the abstract. It also does not describe the outcomes of studies on patient populations, but only mentions that the paradigm has been applied to them. The paper neither follows any kind of research thesis, nor does it give any conclusions, not even in a technological, practical sense. I cannot imagine any readership for which such a paper would be of use.

- The manuscript is riddled with errors in grammar, syntax, and vocabulary. There are three mistakes in the title alone. Indeed, the manuscript gives the impression that none of the five authors ever thoroughly proofread it.

MINOR POINTS

Redundant, given the major flaws of the paper.

Reviewer 2 Report

The authors wrote a literature review on the application of the radial arm maze (RAM) task in real and virtual environments. The focus was on studies with human participants across development. The authors emphasize the potential of the RAM task to study spatial abilities in real and virtual environments. This was an interesting and important overview on human studies using the RAM task which I enjoyed reading. Nevertheless, there are a few points that need attention.

  1. Since the focus is on studies with human participants and studies with animals were excluded, it would be helpful to include this in the title of the manuscript.
  2. It would be helpful to have more details about the eligibility criteria for the selected studies. These are only shortly described in lines 144-146. It is unclear why the last 16 studies were excluded.
  3. The authors should consider to include the number of participants and if possible, the effect sizes of the studies in their literature tables (tables 1 and 2). This would help to estimate the power of the selected studies.

Minor comments:

  1. Line 145: “…some other studies have been excluded because used the RAM…” should probably be ….because they used the RAM…
  2. Line 197 “free-choice3” is unclear
  3. Line 214: It is unclear why a table version of the RAM task allows to study the processing of body–objects interactions. It would be helpful to have more details here.
  4. Lines 218-221: Here is unclear why the table version should elicit stronger processing and induce a powerful multisensory activation [56], inducing an integration of perceptive, motor, and cognitive processes. I think this is more the case when participants are inside the maze and can walk through. The authors should explain this in more detail.
  5. Line 228: Here is an error “O’connor and Glassman RB”

Reviewer 3 Report

Summary:

The authors introduce the use of VR in the study of spatial cognition. They present the two main instruments, initially employed in animal studies (the MWM and the RAM) and then focus their study on the adaptation of RAM in human research through the use of virtual reality technology. They describe pros and cons of such approach and provide a comprehensive review of the studies that have used the virtual version of RAM.

Broad comments:

The topic of this work is of relevance and interest nowadays, as VR is a promising tool in psychology and neuroscience research. Particularly, the development of virtual versions of spatial memory and learning tasks will hep to deal with the increasing incidence of neurodegenerative diseases. Thus, new methodologies to assess and rehabilitate cognitive functions, in particular spatial memory are needed in the immediate future. Nevertheles, I have some concerns about the manuscript I describe below:

Specific comments:

In the lines 111-112, the authors state "spatial working memory component that is not present in real and virtual pool [35]. It is true that the typical reference memory task in the MWM is not considered to include a working memory component. However, the authors should modulate this statement, as it has been proved that spatial memory performance in a virtual navigation task is related to working memory capacity (DOI: 10.3390/brainsci10080552). I just would say that RAM provides more opportunities to assess spatial working memory compared to the real and virtual pool.

Line 118, the authors  state "... apparatuses including both MWM and RAM tasks [i.e., 41]." The authors should mention other experimental procedures employed to assess spatial memory through VR tasks like DOI: 10.1371/journal.pone.0204995. 

 Line 197, there is a typo "free-choice3".

Lines 203-205. The statement "the exploration of an environment through moving in it accelerates the learning process... neurorehabilitation training". Some references supporting this are required.

I do not understand which criteria followed the authors to organize the studies included in tables 1 and 2. There is not a clear rationale for ranging the references and when a particular reference is cited in the text it is difficult to find it in the table.

Probably, the studies cited in tables 1 and 2 should be grouped somehow, for instand according to a factor like "Healthy/Clinical target". The column "target" describes the age of the study sample. Perhaps, this information could be included in the "Healthy/Clinical target" column, for instance stating  "Healthy children" or "Healthy adults"...

Regarding sections 5-7:

Section 5.1. The authors focus on Free-choice and forced-choice version. They describe and discuss the implications of both experimental strategies. As shown in the tables, all the studies revised used one or another strategy, and many used both. I miss the authors analyze and discuss if any of these strategies were more informative, provided different results... according to the sample characteristics. Some information is given in section 6 but superficially.

Section 7 addresses the use of the virtual RAM. The authors cite studies included in table 2. Comparing the information of the "Paradigm" column of table 1 and 2, I see that in table 2 there is majority of studies using a forced-choice paradigm in the virtual RAM. In the case of table 1 there was not such preference. The authors should discuss why most of the studies have used a forced choice paradigm in the virtual RAM.

Section 8:

Lines 335-337. The authors state "... the subject can perform the task exploring the virtual environment in safety, as well the experimenter..." This not characterstic of non-immersive as it also applies immersive VR.  

In this section the authors should mention that other virtual spatial tasks, apart from the virtual RAM, are also being used (like the virtual MWM or the boxes room task) and new research must be conducted to find which procedure is more useful for the assessment and rehabilitation of spatial memory in different pathologies.

Round 2

Reviewer 1 Report

The authors have made a very few changes and corrections to their manuscript, but those are all superficial. Nowhere do they address the major concerns I had about the manuscript.

Reviewer 3 Report

The authors have made some changes following to the feedback previously provided. However, there are some relevant points that have not been addressed properly in the revised version. 

My point regarding section 5.1: "I miss the authors analyze and discuss if any of these
strategies were more informative, provided different results... according to the sample
characteristics. Some information is given in section 6 but superficially".

Some small changes have been made in this section, but not what I demanded. The authors should do a deeper analysis about the use of different strategies and their results depending on the characteristics of the samples. This requires major changes in the manuscript.

Then, this synthesis of reviewed literature should be reflected in the conclusions.
